# (GIGA)byte

DATA RELEASE

# Low-coverage whole genome sequencing for a highly selective cohort of severe COVID-19 patients

Renato Santos[1], Víctor Moreno-Torres[2], Ilduara Pintos[2], Octavio Corral[3], Carmen de Mendoza[2], Vicente Soriano[3] and Manuel Corpas[4],*

1 National Heart & Lung Institute, Imperial College London, London, UK
2 Puerta de Hierro University Hospital & Research Institute, Majadahonda, Madrid, Spain
3 Health Sciences School & Medical Centre, Universidad Internacional La Rioja (UNIR), Madrid, Spain
4 School of Life Sciences, University of Westminster, London, UK

## ABSTRACT

Despite the advances in genetic marker identification associated with severe COVID-19, the full genetic characterisation of the disease remains elusive. This study explores imputation in low-coverage whole genome sequencing for a severe COVID-19 patient cohort. We generated a dataset of 79 imputed variant call format files using the GLIMPSE1 tool, each containing an average of 9.5 million single nucleotide variants. Validation revealed a high imputation accuracy (squared Pearson correlation ≈0.97) across sequencing platforms, showcasing GLIMPSE1's ability to confidently impute variants with minor allele frequencies as low as 2% in individuals with Spanish ancestry. We carried out a comprehensive analysis of the patient cohort, examining hospitalisation and intensive care utilisation, sex and age-based differences, and clinical phenotypes using a standardised set of medical terms developed to characterise severe COVID-19 symptoms. The methods and findings presented here can be leveraged for future genomic projects to gain vital insights into health challenges like COVID-19.

**Subjects** Genetics and Genomics, Bioinformatics, Personalized Medicine

**Submitted:** 25 September 2023

* Corresponding author. E-mail: m.corpas@westminster.ac.uk

Preprint submitted at https://doi.org/10.1101/2024.01.28.577610

## CONTEXT

Coronavirus disease 2019 (COVID-19) caused by the severe acute respiratory syndrome coronavirus 2 (SARS-CoV-2) first appeared in Wuhan, China at the end of 2019 [1]. Clinical presentation of COVID-19 can largely vary, ranging from asymptomatic infections to more severe forms with pneumonia, multiple organ complications, and sepsis [2]. Previous genome-wide association studies (GWAS) have collectively identified an association between genetic risk factors at multiple loci across the human genome and severity and susceptibility to COVID-19 infection [3–5]. Certain patient characteristics, such as older age and male sex, alongside comorbidities such as obesity and cancer, have been shown to contribute to severe outcomes in COVID-19 patients [6]. These existing insights into the ailment have paved the way for new opportunities to explore the determinants of COVID-19 severity [7], particularly due to its potential applications in risk prediction, preventive medicine, and patient management.

Traditionally, genotyping for GWAS has heavily utilised microarray technologies due to their low costs and rapid turnaround times, making them valuable and affordable tools for the high-throughput generation of genomic data [8]. However, arrays are limited by their

experimental design, leading to biases in the data generated. In particular, the prior selection of genetic markers and probes creates an ascertainment bias often resulting in the overrepresentation of intensively researched populations that are more likely to be involved in array development [9]. This contrasts with high-coverage whole genome sequencing (WGS), which has promised the ability to probe variation across the entire human genome, free from the ascertainment bias. This has led to its adoption in large-scale population-level projects [10].

Despite significant cost reductions over time [11], WGS at the clinically accepted standard of 30× coverage [12] remains expensive for many projects, especially for those involving large sample cohorts, similar to those required for GWAS. However, recent studies have demonstrated that sequencing larger numbers of individuals at lower coverages and prioritising cost and haplotype diversity over sequencing depth can yield more allelic information at the cohort and population levels [13]. As a result, low-coverage WGS (lcWGS) has emerged as a cost-efficient alternative to high-coverage WGS, surpassing microarrays in the discovery of common and low-frequency variation [14, 15], particularly in underrepresented populations [16].

Similar to microarrays, lcWGS data can also be imputed using reference panels to enhance resolution and statistical power while maintaining low sequencing and data processing costs. The fundamental principle underlying the genotype imputation algorithms is known as identity-by-descent (IBD), wherein two seemingly unrelated individuals may share segments shorter than 10 centimorgans (cM) inherited from a distant common ancestor [17]. Accordingly, the genotype imputation algorithms compare the sparsely distributed haplotypes present in the lcWGS data with the haplotypes in the high-coverage reference panel to infer genotype likelihoods in the regions not covered by sequencing [18].

Previous imputation methods for lcWGS data exhibited significant drawbacks such as higher costs, longer running time when using large reference panels due to computational complexity [19], and use of more efficient approximations which resulted in lower imputation accuracy [20, 21]. To address these challenges and improve the effectiveness of the tool, we utilised the GLIMPSE1 algorithm [16], a less resource-intensive tool that produces more accurate imputed data than its predecessors, to generate a VCF dataset containing 79 imputed lcWGS samples.

Although genotype imputation in lcWGS datasets shows promise, its practical applications are still in the initial stages. With continuous advancement in sequencing technologies, we expect that imputation methods will play a significant role in unravelling the complexities of the human genome and accelerating discoveries in precision medicine and personalised healthcare.

## DATASET DESCRIPTION

We generated a dataset consisting of 79 VCF files, and their respective FASTQ and CRAM files, using the GLIMPSE1 imputation algorithm [16]. We leveraged the 1000 Genomes Project Phase 3 dataset [22] as the reference panel for haplotypes. Collectively, the dataset comprises approximately 325 GB of FASTQ data, 156 GB of CRAM data, and 6 GB of VCF data.

Our samples were specifically derived from sequenced DNA from a highly selective cohort of patients, mostly made up of Iberian Populations in Spain (IBS) individuals alongside individuals from other genetic backgrounds. All the patients included in the

dataset presented with severe COVID-19 symptoms during the initial wave of the SARS-CoV-2 pandemic in Madrid, Spain.

The dataset a range of 5.01 million to 9.60 million high-confidence single nucleotide variants per VCF file (Figure 1A). The average density of SNVs in the VCF files varied from 0 variants per megabase, in the centromeres and other hard-to-sequence regions, to more than 20,000 variants per megabase (Figure 1B). Most samples in the dataset presented with a variant overlap higher than 98% (Figure 1C). The high variant overlap was an expected outcome of imputation against a single reference panel [23] with the lower overlap levels of two samples against the rest of the dataset which was explained by the lower number of variants in the files. To facilitate access for researchers interested in further studying this data, it has been made available for reuse in the European Genome-phenome Archive [24], under the accession number EGAS00001007573.

## PATIENT COHORT CHARACTERISATION

We examined the dataset focusing on three main aspects: (1) a general characterisation of the patients by age, sex, and ancestry; (2) hospital stays and time spent in the intensive care unit (ICU); and (3) the distinct clinical phenotypes presented by the patients (respiratory, thromboembolic, cardiovascular, etc.). Further details on the patient's demographic information and clinical history can be found in Supplementary File 1 [25].

### Sampling strategy

The 79 genomic samples analysed in this study constituted a subset of a larger cohort of individuals whose exomes were initially sequenced and analysed as part of a comprehensive investigation into genetic determinants for COVID-19 severity [26]. The subset was selected based on the quality assessment of DNA samples suitable for PCR-free library preparation for lcWGS. All the individuals were patients hospitalised between March and June 2020, during the first wave of the SARS-CoV-2 pandemic in Spain, at a tertiary referral hospital in Madrid, and confirmed to be infected with SARS-CoV-2. We aimed to select patients with the following clinical profile: (1) younger than 60 years old; (2) experienced fever and respiratory symptoms for more than three days; (3) blood oxygen saturation level below 93%; (4) bilateral pneumonia on imaging tests; and (5) no comorbidities, such as diabetes, obesity, or immunosuppressive conditions. At the time of recruitment of the study, the vaccines were yet to be developed.

### Demographic characterisation

Through our analysis, we aimed to create a comprehensive demographic profile of our cohort of severe COVID-19 patients. The age distribution in the cohort (Figure 2A) was characterised by a distinct right skew, with a higher prevalence of individuals falling within the 45–64 age bracket and particularly concentrated around 55–59 years, which aligned with the latest understanding of the correlations between older age and severe COVID-19 outcomes [6]. However, the lower tail-end of the distribution also underscored the fact that severe COVID-19 was not strictly age-related and young individuals might also suffer from severe manifestations of the infection.

The sex distribution (Figure 2B) showed a higher frequency of male patients compared to females. This finding concurred with previous research indicating that men were at a higher risk of developing severe COVID-19 [6]. Investigating the age distribution in relation



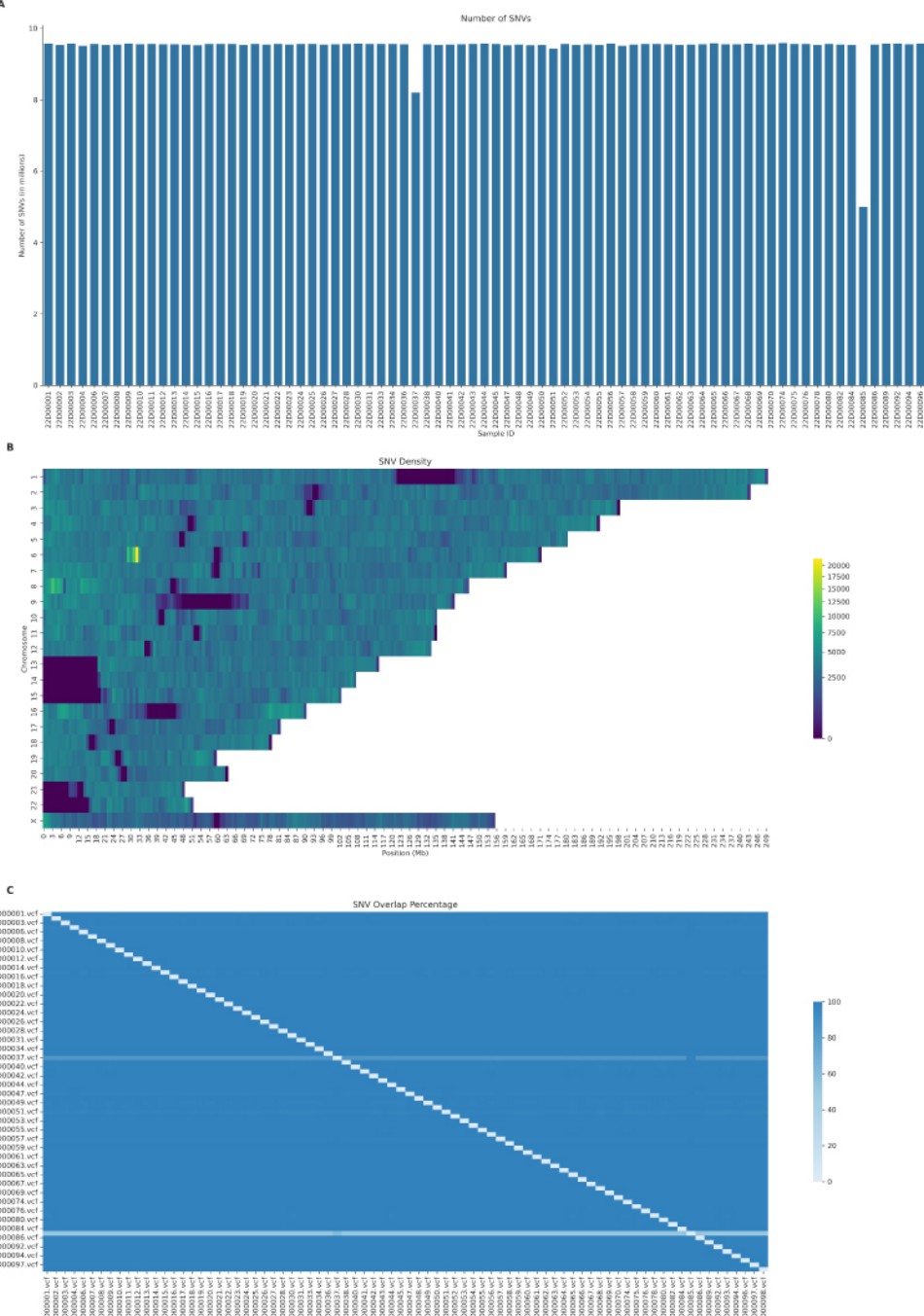

**Figure 1.  Characterisation of the genomic landscape of the imputed VCF dataset.**
(A) The number of high-confidence single nucleotide variants (SNVs) for the 79 VCF files in the severe COVID-19 dataset. The *x*-axis represents the sample IDs in the dataset, while the *y*-axis denotes the total counts of SNVs for each sample in millions ($1 \times 10^6$). (B) SNV density across chromosomes in the dataset. The heatmap shows the distribution of SNVs along the chromosomes, with each row representing a chromosome, 1–22 and ×, and each column a bin-sized 1 megabase (Mb). The number of SNVs in each bin is weighted for the number of samples containing each variant, to represent an average sample in the dataset. Colours range from low (blue) to high (yellow) SNP density. (C) Percentage of overlap of SNVs between samples. The heatmap visualises the extent of shared SNVs across different samples, with each cell representing the overlap percentage from the sample on the *x*-axis to the sample on the *y*-axis. Therefore, the percentage value shown is the proportion of SNVs in the sample on the *x*-axis and also found in the sample on the *y*-axis. The colour gradient from light to dark blue signifies an increasing percentage of overlap.

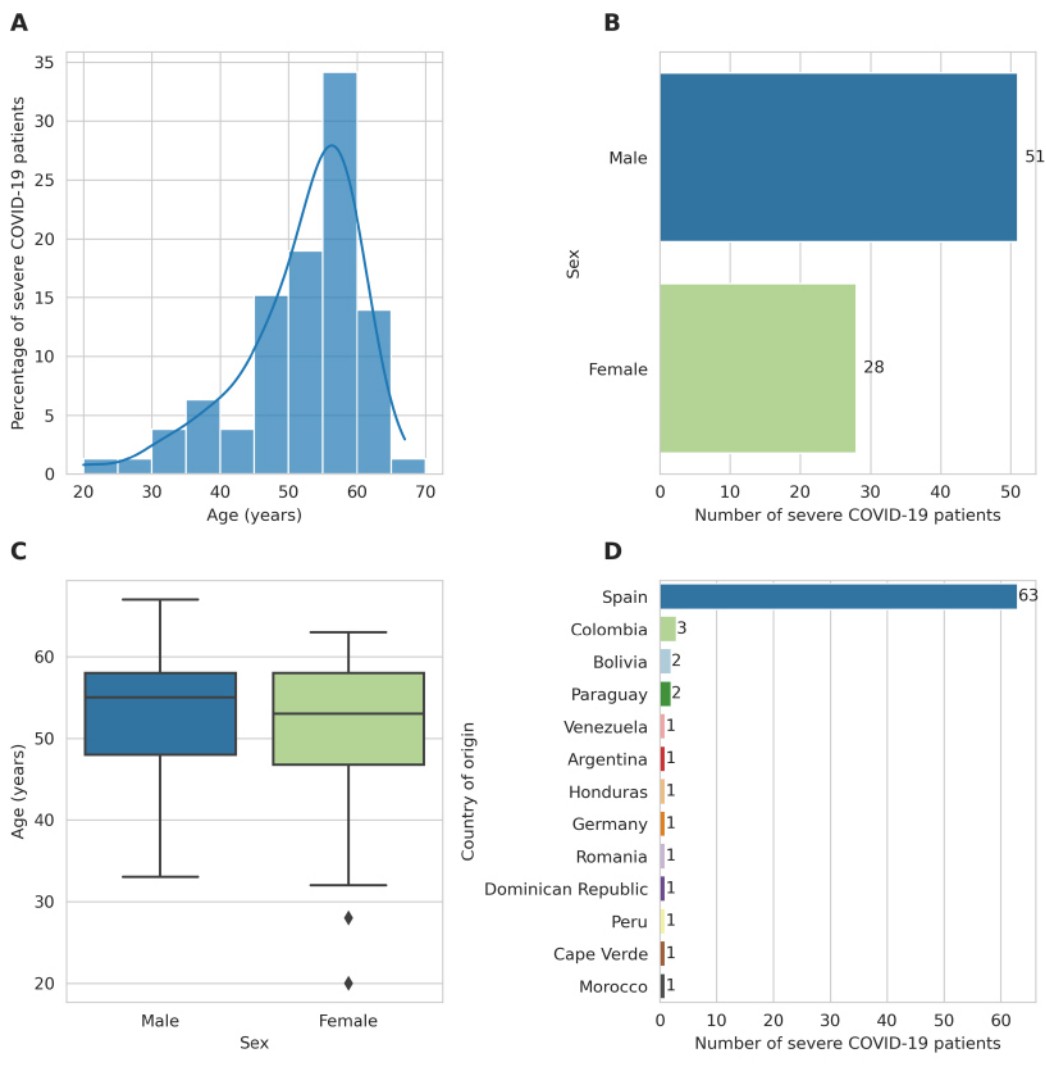

**Figure 2.  Demographic and geographic characterisation of the severe COVID-19 patient cohort.**
(A) Distribution of patient ages with severe COVID-19 cases in our cohort. Each bar signifies an age bracket comprising 5-year increments, with its height denoting the proportion of individuals within that age range. The plot is overlaid with a Kernel Density Estimation (KDE) curve, which provides a smoothed estimation of the age distribution. (B) Patients' stratification by sex. Each bar represents one sex, with its length indicating the number of patients belonging to that sex. (C) Distribution of patient age by sex. The boxplot presents the age distribution for each sex. Each box represents the interquartile range (IQR) of ages for either males or females, with the dividing line representing the median age. The diamonds represent outliers. (D) Distribution of patients by country of origin. Each bar corresponds to a country, and its length indicates the number of patients from that country.

to sex (Figure 2C), indicated that both males and females had a similar median age of 55 and 53 years, respectively. However, the female age distribution demonstrated a broader range (20–63 years) and slightly higher variability (SD 10.74 years) compared to the male distribution, which exhibited a relatively narrower range (33–67 years) and lower variability (SD 7.35 years). Thus suggesting a wider and more variable range of age-related risk among females in the cohort.

Lastly, studying the country of origin of the patients (Figure 2D) revealed that most of the patients in the cohort originated from Latin American countries and Spain. This

geographical distribution is mostly reflected in the demographic composition of Madrid, Spain, where the samples were collected. To expand our analysis beyond demographics and understand the genetic makeup of our cohort, we also performed a Principal Component Analysis (PCA) on our 79 samples, post-imputation and variant filtering, against the backdrops of the 1000 Genomes Project [22] global superpopulations and IBS population.

The global PCA plots (Figure 3A) indicated that most samples clustered within the European (EUR) group which mirrored the fact that a significant proportion of our cohort hailed from Spain. Additionally, a subset of patients was found within or near the Admixed American (AMR) and South Asian (SAS) clusters, reflecting the Latin American patients in our cohort, and the mixed ancestry common in Latin American populations. A few patients also clustered within the African (AFR) group, presumably representing the African ancestry of patients hailing from Cape Verde and Morocco.

In the IBS-specific PCA plots (Figure 3B), most severe COVID-19 patients form a distinct clustered close to the 1000 Genomes IBS population, indicating a shared genetic background with this group, likely representing the individuals with IBS ancestry. However, it is worth noting that subtle regional genetic variations within the Iberian population could contribute to the observed dispersion within this shared genetic background, particularly along the third principal component. The figure also showed a dispersion of patients alongside the left side of the plots, representing individuals with ancestries other than IBS. These outcomes highlighted the genetic diversity in our cohort, owing to the inclusion of patients from Spain and other nations.

## Hospital stays

Examining the hospital medical records of the patients provided valuable insights into the hospitalisation experience of individuals with severe COVID-19. By examining these trends, we gained a better understanding of the potential sex and age-based differences in the duration of hospitalisation and the level of care required.

Firstly, we analysed the distribution of hospitalisation days in our patient cohort (Figure 4A). The distribution was notably skewed to the right, with most patients requiring relatively short hospital stays between 1 and 34 days. However, the distribution's right tail presented a subset of patients who required significantly longer stays of up to 202 days. This could be attributed to cases of COVID-19 with increased severity which necessitated additional medical attention.

Furthermore, an evaluation of the distribution of hospital stays by sex (Figure 4B) revealed that the median duration of hospital stays was similar for both sexes. Nevertheless, the distribution for male patients exhibited greater variability, heightened by the presence of some outliers who spent an unusually high number of days in the hospital, representing severe or complex cases that required a significantly longer time for recovery and medical management. This could mean that the severe disease progression and recovery time in males was less consistent than in females, possibly due to a wider range of severity in clinical presentations among male patients.

In addition, we investigated the use of the intensive care unit (ICU). Approximately 25% of the cohort was admitted to the ICU during their hospitalisation (Figure 4C), indicating that, despite the severity of their COVID-19 symptoms, most patients were managed without the need for intensive care. However, a much larger proportion of males necessitated ICU

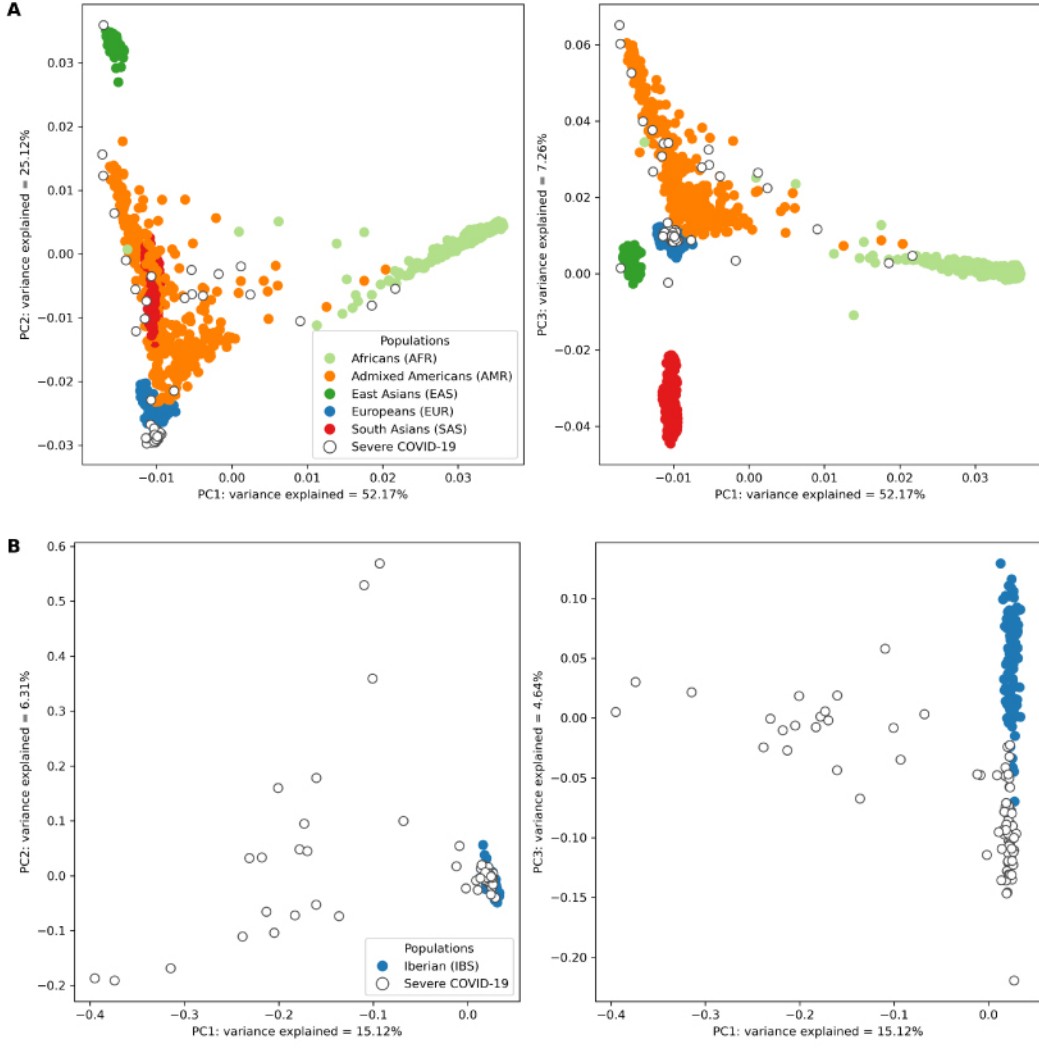

**Figure 3.  Principal component analysis of genetic variation in the severe COVID-19 patient cohort against the 1000 Genomes Project global superpopulations and IBS (Iberian Populations in Spain) population.**
(A) Projection of imputed low-coverage whole-genome sequencing (lcWGS) data from severe COVID-19 patients against the backdrop of global superpopulations from the 1000 Genomes Project. Each point represents an individual, colour-coded according to their superpopulation. Severe COVID-19 patients are distinguished by points with a white fill and coloured border. The *x*-axis and *y*-axis on the two subplots represent the first and second, and first and third principal components, respectively. The percentage of variance is explained by each component indicated in the axis label. (B) Focused view of the genetic variation within the Iberian (IBS) population and the severe COVID-19 patients. Individuals from the IBS population are represented by solid-coloured points, while those with severe COVID-19 are represented by points with a white fill and coloured border. The *x*-axis and *y*-axis on the two subplots represent the first and second, and first and third principal components, respectively, with the percentage of variance explained by each component indicated in the axis label.

admission than females (Figure 4D). This further reinforced findings from numerous studies that also identified male sex as a risk factor for severe COVID-19 outcomes [6].

We further stratified the ICU data by patient age (Figure 4E), showing that the majority of patients who were admitted to the ICU were between 45 and 70 years old, which underscored the heightened risk of severe outcomes in these age groups. Finally, we investigated the ICU stay duration among those requiring such care (Figure 4F). Reflecting the duration of hospital stays for the cohort, majority of patients admitted to the ICU spent

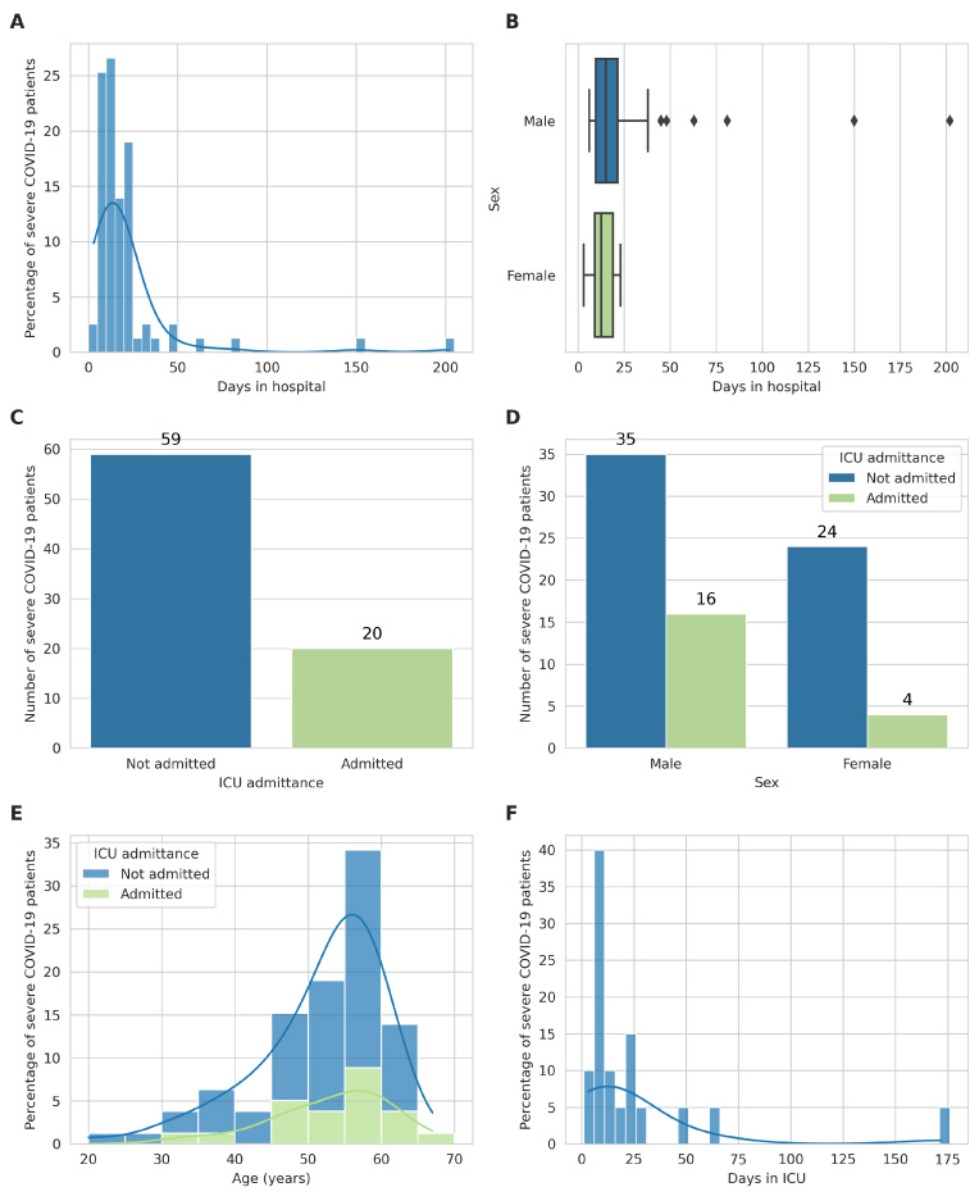

**Figure 4. Analysis of hospital stays among the severe COVID-19 patient cohort.**
(A) Distribution of hospital stay durations in our cohort. Each bar corresponds to an interval of 5 days of stay at the hospital, with its height indicating the proportion of patients'stay duration within that duration interval. The plot is overlaid with a Kernel Density Estimation (KDE) curve which provides a smoothed estimate of the duration distribution. (B) Stratification of hospital stay durations by sex. This boxplot presents the distribution of hospital stays for each sex. Each box represents the interquartile range (IQR) of the duration of hospital stays for one sex, with the line inside the box marking the median duration. The diamonds represent outliers. (C) Distribution of patients admitted to the Intensive Care Unit (ICU). Each bar corresponds to either patients admitted to the ICU (green) or patients not admitted to the ICU (blue), with its height indicating the number of patients in each category. (D) Distribution of patients admitted to the ICU by sex. Each pair of bars corresponds to one sex, with their height indicating the proportion of patients of that sex admitted to the ICU. Each bar corresponds to either patients admitted to the ICU (green) or patients not admitted to the ICU (blue) and the bar's height indicating the number of patients in that category. (E) Distribution of ages of patients admitted to the ICU. Each bar corresponds to an age group of 5 years, with the height indicating the proportion of patients in that age group. The plot is overlaid with a KDE curve, which provides a smoothed estimate of the age distribution. (F) Distribution of ICU stay durations among patients admitted to the ICU. Each bar corresponds to an interval of ICU stay durations of 5 days, with its height indicating the number of patients within that duration interval. The plot is overlaid with a KDE curve, that provides a smoothed estimate of the duration distribution. Only patients who were admitted to the ICU are represented in this plot.

between 1 and 35 days there. However, a considerable subset of patients experienced significantly longer ICU stays, representing a wide spectrum of disease severity and recovery rates within the critical care cohort.

## Clinical phenotypes

An analysis of the phenotypes of the severe COVID-19 patient cohort revealed valuable insights into the most common phenotypes associated with severe forms of the disease, their frequency and relationships. While established that COVID-19 phenotype ontologies were readily available [27, 28] they lacked the level of granularity we required to comprehensively characterise the clinical phenotypes of our cohort. Therefore, we devised a specialised set of standardised terminology comprising 28 medical terms that were organised into 4 primary clinical categories: Pulmonary, Extra-Pulmonary, Coagulation, and Systemic phenotypes. Subsequently, we evaluated each patient's record for the presence of these terms.

Table 1 provides a detailed breakdown of the number of patients associated with each specific phenotype, within the four major clinical categories. We found that the Pulmonary category, which includes pneumonia, ARDS (acute respiratory distress syndrome), and a combination of ARDS with admission to the ICU, was the most prevalent among our cohort. Indeed, pneumonia alone was identified in 78 patients. The Extra-Pulmonary category covered a broad range of clinical symptoms and conditions, with liver hepatitis and gastrointestinal diarrhoea being the most common as observed in 10 patients each. The Coagulation category focused on thrombotic events and related conditions. Pulmonary embolism and deep venous thrombosis, each identified in 5 patients, were most prevalent. Finally, the Systemic category, pertaining to conditions that affect the patient's overall health and well-being, such as persistent fever and symptoms like fatigue and headache. Persistent fever was the most common Systemic phenotype, observed in 33 patients.

To further investigate the relationships between the phenotypes in our patient cohort and to determine whether any of them were likely to co-occur, we performed a Spearman correlation analysis using the function *corr(method="spearman")* from the seaborn package for Python [29], and visualised the results in a heatmap (Figure 5). These correlations suggested that the patients with one of these phenotypes were more likely to show others, pointing to possible common underlying pathways or simultaneous occurrence in severe disease presentation.

The plot indicated that most phenotypes are not strongly correlated, hence, the presence of one phenotype does not necessarily predict the presence of another. This could be indicative of the diverse clinical manifestations of severe COVID-19 with different phenotypes appearing independently among patients. However, there were several pairs of phenotypes exhibiting higher degrees of correlation. This became particularly evident in neurological conditions, such as the correlations between psychiatric disorders, encephalopathies and polyneuropathies, which appeared to be correlated to a relatively high degree. In addition, moderate correlations were shown between the former three neurological phenotypical categories and exanthema, myopathies, and bone marrow abnormalities. Finally, some moderate correlations were observed between the ARDS & ICU phenotype and a few other phenotypes thus pointing to the additional occurrence of various phenotypes of COVID-19 severity in patients admitted to the ICU with ARDS.

**Table 1.** Frequency of severe COVID-19 phenotypes in the patient cohort.

| Major Phenotypes | Phenotype | Phenotype ID | Patient count n = 79 |
|---|---|---|---|
| Pulmonary n = 78 | Pneumonia | V-1 | 78 |
| | ARDS (acute respiratory distress syndrome) | V-2 | 47 |
| | ARDS & ICU | V-3 | 20 |
| Extra-pulmonary n = 37 | Skin – exanthema | V-4 | 4 |
| | Heart – myocarditis | V-5 | 1 |
| | Heart – arrhythmia | V-6 | 4 |
| | Liver – hepatitis | V-7 | 10 |
| | Kidney – glomerulonephritis | V-8 | 0 |
| | Kidney – tubulopathy | V-9 | 4 |
| | Neurological – encephalitis/encephalopathy | V-10 | 6 |
| | Neurological – psychiatric (delirium, etc.) | V-11 | 6 |
| | Neurological – polyneuropathy (neuropathy, Guillain-Barré, etc.) | V-12 | 6 |
| | Neurological – myelitis | V-13 | 0 |
| | Neurological – seizure | V-14 | 0 |
| | Gastrointestinal – diarrhoea | V-15 | 10 |
| | Gastrointestinal – nausea/vomiting | V-16 | 3 |
| | Endocrine dysfunction (thyroid, etc.) | V-17 | 2 |
| | Musculoskeletal – myopathy | V-18 | 1 |
| | Musculoskeletal – arthritis | V-19 | 0 |
| | Bone marrow – blood cytopenia, pancytopenia/aplasia | V-20 | 5 |
| Coagulation n = 12 | Pulmonary embolism | V-21 | 5 |
| | Deep venous thrombosis | V-22 | 5 |
| | Peripheral arterial thrombosis | V-23 | 0 |
| | Stroke | V-24 | 0 |
| | Ischemic heart event | V-25 | 1 |
| | Disseminated intravascular coagulation | V-26 | 3 |
| Systemic n = 41 | Persistent fever | V-27 | 33 |
| | Fatigue, malaise, headache, arthromyalgias | V-28 | 20 |

This exploratory analysis highlighted the diverse ways in which manifestations of severe COVID-19 could call attention to the crucial role played by comprehensive and nuanced clinical phenotyping in improving our understanding and management of the disease.

The table presents the distribution of our 28 severe COVID-19-specific phenotypes organised into four major clinical categories: Pulmonary, Extra-pulmonary, Coagulation, and Systemic, as observed in our severe COVID-19 patient cohort. For each category, the total number of unique patients with at least one phenotype in the category is indicated. Each phenotype is listed with a unique Phenotype ID (V-1 to V-28) and the number of patients who were identified with that phenotype.

The plot illustrates the Spearman correlations between our 28 severe COVID-19-specific phenotypes. Each square in the heatmap represents the correlation between two phenotypes, with the colour of the square indicating the strength of the correlation, and the number inside each square represents the correlation coefficient. Only statistically significant ($p < 0.05$) correlation coefficients are shown. Darker colours represent stronger positive or negative correlations here red represents positive correlations and blue represents negative correlations. These indicate that individuals displaying one phenotype are more or less likely to exhibit the other phenotype as well, suggesting potential underlying mechanisms for the progression of severe COVID-19.

## METHODS

## DNA extraction and library preparation

We collected blood samples for each patient in 10 mL EDTA tubes. We then centrifuged the tubes at 3000 rpm for 10 minutes to isolate the buffy coat, which we subsequently froze to −20 °C until further use. Afterwards, we used the Maxwell RSC Buffy Coat DNA Kit (AS1540,

Heatmap of Phenotype Correlations

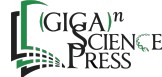

**Figure 5.** **Heatmap of phenotype correlations in the severe COVID-19 patient cohort.**

Promega UK, Southampton, United Kingdom) to isolate genomic DNA from frozen buffy coat samples. We assessed the concentration of the genomic DNA using spectrometric analysis. We then fragmented the DNA using a Covaris E220 focused ultrasonicator (Covaris Ltd., Brighton, United Kingdom) to generate 350 bp length DNA fragments. The following parameters were used for the fragmentation process: *6 cycles, PIP 75, Cycles/Burst 1000, Duty Factor 20%, Duration 20s*. Post fragmentation, we prepared DNA libraries using the MGIEasy PCR-Free DNA Library Prep Set (1000013453; MGI Tech Co. Ltd, Shenzhen, China). The concentration of the DNA libraries was assessed using a Qubit 3.0 fluorometer (Life Technologies). Finally, we sequenced the libraries on an MGI DNBSEQ-G400 sequencer (RRID:SCR_017980; MGI Tech Co. Ltd), with a target sequencing depth of 1×.

(GIGA)ⁿ
SCIENCE
PRESS

### Sequencing quality control and preprocessing

We performed quality control and preprocessing of the resulting FASTQ files using the nf-core [30] Sarek pipeline v3.1.2 [31–40]. The following parameters were applied during the pipeline execution: *nextflow run nf-core/sarek -r 3.1.2 -profile docker --input samplesheet.csv --outdir /mnt/e/Sarek/out/ --trim_fastq --igenomes_base /mnt/e/Sarek/references --genome GATK.GRCh37 --skip_tools strelka --seq_platform 'MGI'*. We used the recalibrated base quality scores CRAM files produced by the Sarek pipeline as the input for the subsequent step.

### Imputation using GLIMPSE

We calculated genotype likelihoods using bcftools mpileup v1.16 (RRID:SCR_005227) [40], with the parameters *-I -E -a 'FORMAT/DP'*, followed by genotype calling using bcftools call, with parameters *--ploidy GRCh37 -S ploidy.txt -Aim -C alleles*. The file 'ploidy.txt' contained information about the sex of each sample, which was necessary to generate correct genotype calls for chromosome × in males [16].

Following which we imputed the low-coverage genomes with GLIMPSE v1.1.1 [16]. Firstly, we split each chromosome into 2 Mb chunks, with 200 kb buffer regions on each side of a chunk using GLIMPSE_chunk with the parameters *--window-size 2000000 --buffer-size 200000*. Secondly, we used GLIMPSE_phase to impute each chunk with the tool's default iteration parameters. GLIMPSE_phase imputation was multithreaded with GNU Parallel v20230522 [41]. We used the 1000 Genomes Phase 3 dataset [22] as the reference panel, since it had shown that inclusion of diversity in reference panels improves the quality of imputation by reducing missing genotype calls [42, 43]. Finally, we used GLIMPSE_ligate to join the imputed chunks into entire chromosomes, followed by bcftools concat to merge all chromosomes into a single VCF file, containing chromosomes 1 to 22 and ×, for each sample.

### Post-imputation filtering

Following imputation, we filtered the VCF files, to prioritise the most reliable genotype calls for further analysis. We only proceeded with minor allele frequency (MAF) above 2% and maximum genotype probability (GP) above 80%. We determined the MAF value through our validation process (see Data Validation) as the minimum threshold of acceptable imputation accuracy of $r^2 \cong 0.9$. The GP field represented the likelihood of each genotype being accurate and was expressed as a value between 0 and 1 with the sum of probabilities across all possible genotypes totalling 1 [44]. The chosen cutoff was determined as the best compromise between imputation accuracy and loss of information [44]. With this approach, we generated a dataset of imputed VCF files, from the DNA samples of our severe COVID-19 cohort.

### Principal component analysis

We performed a principal component analysis (PCA) to assess the genetic ancestries of our patient cohort, using PLINK v.1.90b6.21 (RRID:SCR_001757) [45]. To accomplish that, firstly we normalised the variants from chromosomes 1 to 22 of the 1000 Genomes Project Phase 3 dataset [22] using bcftools [40]. We then split the multi-allelic calls and left-aligned indels against the reference genome using bcftools norm, with the parameters *-m-any --check-ref w*; followed by bcftools annotation with the parameters *-x ID -I +'%CHROM:%POS:%REF:%ALT'*, to normalise the naming of unset IDs; and bcftools norm to

remove duplicate records using the parameters *-Ob --rm-dup both*. Furthermore, we filtered this global variant dataset to create a distinct subset containing variants exclusively from the 1000 Genomes Project IBS individuals. We used bcftools view, with the *-S* parameter, which included only variants originating from the specified 1000 Genomes IBS sample IDs.

We then converted the two datasets, 1000 Genomes global and IBS, to binary PLINK format, using PLINK with the parameters *--keep-allele-order --vcf-idspace-to _ --const-fid --allow-extra-chr 0 --split-x b37 no-fail --make-bed*. Next, we used PLINK to identify population-specific markers by filtering the variants within both datasets based on MAF and variance inflation factor (VIF), using the parameters *--maf 0.10 --indep 50 5 1.5*. We then pruned the datasets through *–extract* to include only the variants that fulfilled the thresholds.

Subsequently, we merged the variants from chromosomes 1 to 22 of all 79 post-filtering imputed VCF files into a single BCF file using bcftools merge with the parameters *-r $(seq -s, 1 22) --missing-to-ref*, followed by a normalisation process akin to that of the 1000 Genomes global and IBS datasets. Following this we converted the merged imputed dataset to binary PLINK format.

We used PLINK again to determine common variants between the merged imputed dataset and the populational markers in the 1000 Genomes global and IBS datasets, using *–bmerge*, followed by use of *–extract* for the extraction of common variants. Finally, we merged the extracted variant datasets with *–bmerge* and calculated 20 principal components using *–pca*. We then plotted the first three principal components using the matplotlib package for Python [46].

## DATA VALIDATION

To validate our imputed dataset, we sought to quantify the accuracy of our imputation process and determined whether it is affected by the choice of short-read sequencing platforms. To perform that we obtained a healthy IBS genome (IBS001) independent from our patient cohort, sequenced at 40× coverage using an Illumina system, at Dante Labs (Cambridge, United Kingdom), and an MGI system, at BGI Tech Solutions Hongkong Co Ltd (Tai Po, Hong Kong). After quality control, alignment, and pre-processing, we obtained two CRAM files for the IBS001 individual: *IBS001_illumina* with an average depth per base of 39.34 and *IBS001_bgi* with an average depth per base of 41.64. The average depth per base was calculated using mosdepth v0.3.3 [37], with the *-x* parameter, against the GATK b37 reference genome.

We downsampled both versions of the IBS001 genome to 1× coverage using samtools v1.16 (RRID:SCR_002105) [40]. We calculated the subsampling fractions by dividing the target depth per base (1.00) by their respective average depth per base of each file. Hence, the genome sequenced on the Illumina platform was also downsampled using the command *samtools view -s 0.025419420437214,* and the genome sequenced on the MGI platform was downsampled using *samtools view -s 0.0240153698366955*.

The two resulting low-coverage downsamplings were used for the genotype calling, imputation, and post-imputation filtering steps as described in the *Methods* section. This process resulted in four VCF files: two pre-filtering files, two post-filtering files from the two sequencing platforms, Illumina and MGI. These four files served as the imputed genotypes for the validation process. Additionally, we also performed the genotype calling step on the



**Table 2.** Aggregate GLIMPSE1 imputation concordance for the IBS001 validation genome.

| Coverage | $r^2$ concordance pre-filtering | | $r^2$ concordance post-filtering | |
|---|---|---|---|---|
| | BGI 40× | Illumina 40× | BGI 40× | Illumina 40× |
| BGI 1× | 0.962085 | 0.962852 | 0.970049 | 0.970573 |
| Illumina 1× | 0.961485 | 0.962807 | 0.969826 | 0.970724 |

original high-coverage files, with the VCF output as the gold standard genotypes for validation.

To measure imputation accuracy we used GLIMPSE_concordance [16] to calculate a squared Pearson correlation between the high-coverage and imputed dosages across chromosomes 1 to 22 and ×, within several MAF bins (0–0.02%, 0.02–0.05%, 0.05–0.1%, 0.1–0.2%, 0.2–0.5%, 0.5–1%, 1–2%, 2–5%, 5–10%, 10–15%, 15–20%, 20–30%, 30–40%, and 40–50%), and a single aggregate squared Pearson correlation across all sites. We only used sites in the validation data with a minimum depth of 8 reads and a minimum posterior probability of 0.9999. To accomplish the same, we used the parameters *––minDP 8 ––minPROB 0.9999 ––bins 0.00000 0.0002 0.0005 0.001 0.002 0.005 0.01 0.02 0.05 0.1 0.15 0.2 0.3 0.4 0.5*.

We conducted a concordance assessment by comparing the imputed pre-filtering IBS001 genomes against the high-coverage validation IBS001 genomes, within sequencing platforms (BGI 1× vs BGI 40× and Illumina 1× vs Illumina 40×) and across sequencing platforms (BGI 1× vs Illumina 40× and Illumina 1× vs BGI 40×). This was done to identify any quantifiable differences in imputation quality arising from the use of different short-read sequencing platforms. Across all four platform comparisons (Figure 6A), GLIMPSE accurately imputed variants in the 2–5%, 5–10%, 10–15%, 15–20%, 20–30%, 30–40%, and 40–50% MAF bins, represented by an $r^2$ correlation equal to or higher than 0.90. However, as anticipated, imputation accuracy steadily decreased for MAFs lower than 2%, likely due to the underrepresentation of rare variants in the 1000 Genomes reference panel [16]. Overall, this resulted in an aggregate $r^2$ correlation of approximately 0.96 across all MAF bins for both IBS001 genomes (Table 2).

Furthermore, we evaluated the impact of the filtering process on the dataset accuracy. We repeated the concordance comparison using GLIMPSE_concordance, instead of imputed post-filtering IBS001 genomes against the high-coverage validation IBS001 genomes. Due to the removal of low confidence sites during filtering, the $r^2$ correlation in the 2–5%, 5–10%, 10–15%, 15–20%, 20–30%, 30–40%, and 40–50% MAF bins improved slightly (Figure 6B). In turn, this improved the aggregate $r^2$ correlation to approximately 0.97 on all four platform comparisons we performed (Table 2).

In conclusion, the validation of our imputation and filtering process shows that GLIMPSE1, with the 1000 Genomes Project Phase 3 [22] as the reference panel, can be confidently used to impute variants with MAF up to approximately 2%.

The table displays the aggregate $r^2$ correlation results obtained from the concordance assessment of the imputed IBS001 genomes against the high-coverage validation IBS001 genomes. The analysis was performed for chromosomes 1 to 22 and × within sequencing platforms (BGI 1× vs BGI 40× and Illumina 1× vs Illumina 40×) and across sequencing platforms (BGI 1× vs Illumina 40× and Illumina 1× vs BGI 40×). The table presents the aggregate $r^2$ correlation values, indicating the overall imputation accuracy across all sites.

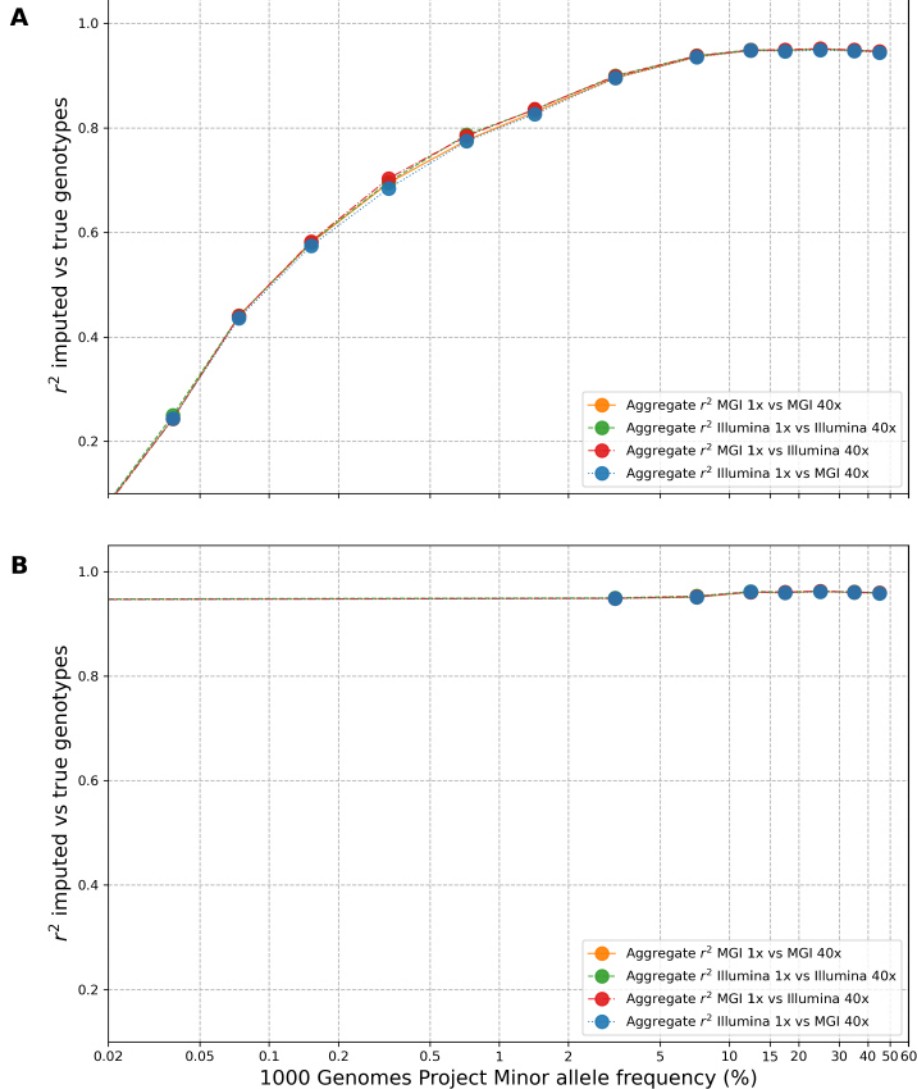

**Figure 6.   Assessment of GLIMPSE1 imputation concordance within different minor allele frequency (MAF) bins for the IBS001 validation genome.**
(A) Squared Pearson correlation ($r^2$) between high-coverage and pre-filtering imputed dosages segregated into various MAF bins. The *x*-axis shows MAF bins, ranging from 0 to 50%, and the *y*-axis shows the squared Pearson correlation coefficient ($r^2$). The analysis was performed for chromosomes 1 to 22 and ×, within sequencing platforms (BGI 1× vs BGI 40× and Illumina 1× vs Illumina 40×) and across sequencing platforms (BGI 1× vs Illumina 40× and Illumina 1× vs BGI 40×). (B) Squared Pearson correlation ($r^2$) between high-coverage and post-filtering imputed dosages segregated into various MAF bins.

## RE-USE POTENTIAL

Despite continuous improvements in genotype imputation algorithms, lcWGS imputation remains underutilised as an economical alternative over higher-coverage sequencing. Additionally, the understanding of host genetic markers that predispose COVID-19 severity is still limited [7].

In this context, our manuscript's dataset, coupled with the innovative strategies employed presents a promising outlook. Not only do we showcase the viability of using



lcWGS imputation to generate data for the study of disease-related genetic markers, but also present a robust validation methodology to ensure the accuracy of the data produced. Our ambition is to inspire confidence and stimulate further interest from researchers who wish to deploy a similar approach to a range of other infectious diseases, genetic disorders, or population-based genetic studies, particularly in large-scale genomic projects and resource-limited settings where sequencing at higher coverages could prove to be prohibitively expensive.

It is important to note, however, that the inherent probabilistic nature of imputed low-coverage genotypes can introduce uncertainty into downstream analyses and measures should be taken to mitigate such errors. For example, Petter and Ding [47] proposed a statistical calibration method for polygenic scores (PGS) based on imputed lcWGS genotypes, which showed improved estimations of PGS over traditional calculations that ignore genotyping errors in low-coverage sequencing. Understanding the nature of genotyping error is essential to accurately interpret imputed lcWGS results. Hence, researchers should be mindful of adopting similar approaches when utilising imputed datasets such as the one presented here.

Beyond the immediate implications in lcWGS imputation, this dataset serves as a valuable resource for investigators studying genetic markers associated with COVID-19 severity. Specifically, the meticulous methodology utilised to characterise our patient cohort, through standardised clinical terminology which paved the way for the discovery of genetic components that might be linked to severe COVID-19 disease manifestations and progressions. The scope of this methodology could be expanded and tailored to analyse hospitalisation trends in other clinical cohorts. Thus serving as a template for future studies aiming to comprehensively characterise complex diseases.

The analysis of patient hospitalisation particularly focused on sex and age-related differences has the potential to inform healthcare policy and clinical guidelines. Similarly, the insights gained from hospital stay distributions, ICU admissions, and the identification of disease severity across different demographics can have broader applications beyond its present use case. For instance, the insights could be translated into personalised care strategies or even underpin predictive models for assisting healthcare providers in delivering more effective treatments [48, 49].

Finally, the validation data regarding comparisons of short-read sequencing platforms is of great importance. As genomic research progresses, the accuracy and reliability of different sequencing platforms become increasingly critical. By offering a comparison of imputation accuracy between Illumina and MGI sequencers, we provide an avenue for other researchers to make informed decisions about their sequencing platforms. This has become especially relevant as the scientific community strives to standardise genetic research methodologies [50] to ensure consistent results and comparable outcomes across different studies.

In conclusion, the dataset presented here, though primarily focused on COVID-19 severity, transcends the scope of this study and can be utilised in multiple domains of scientific research. We encourage its reuse hoping that its integration into other studies will advance our collective understanding and response to complex health challenges, such as those presented by COVID-19.

## AVAILABILITY OF SOURCE CODE AND REQUIREMENTS

- Project name: GLIMPSE low-coverage WGS imputation
- Project home page:
  https://github.com/renatosantos98/GLIMPSE-low-coverage-WGS-imputation
- Operating system(s): Linux
- Programming language: Bash and Python3
- Other requirements: GLIMPSE 1.1.1; samtools 1.16; bcftools 1.16; Python 3.10.8; numpy 1.26.4; matplotlib 3.8.3; pandas 2.2.1; seaborn 0.12.2; parallel 20230522; mosdepth 0.3.3; plink 1.90b6.21; bc 1.07.1.
- License: MIT license.

## DATA AVAILABILITY

The clinical dataset is available in the European Genome-phenome Archive, under the accession number EGAS00001007573. The other datasets (including summarised clinical histories and phenotypes) supporting the results of this article are available in our Figshare collection [51] and GitHub repository [52]. Additional data and code snapshots are also available in the GigaDB repository [53].

## LIST OF ABBREVIATIONS

AFR: 1000 Genomes Africans superpopulation; AMR: 1000 Genomes Admixed Americans superpopulation; ARDS: acute respiratory distress syndrome; cM: centimorgans; COVID-19: coronavirus disease 2019; EDTA: ethylenediaminetetraacetic acid; EUR: 1000 Genomes Europeans superpopulation; GP: genotype probability; GWAS: genome-wide association studies; IBD: identity-by-descent; IBS: 1000 Genomes Iberian Populations in Spain population; ICU: intensive care unit; lcWGS: low-coverage whole genome sequencing; MAF: minor allele frequency; PCA: principal component analysis; PGS: polygenic scores; SAS: 1000 Genomes South Asians superpopulation; SARS-CoV-2: severe acute respiratory syndrome coronavirus 2; VCF: variant call format; WGS: whole genome sequencing.

## DECLARATIONS

### Ethics approval and consent to participate

This study was evaluated and approved by the Clinical Research Ethics Committee of Hospital Clínico San Carlos (code number: 20/313-E_COVID) in Madrid, Spain. In compliance with the provisions of the Declaration of Helsinki and the legislation in force in Spain regarding research with human beings. Patients were informed about their participation in this clinical study, clarifying that their participation was voluntary and did not imply any change in their treatment or medical care compared to what they would receive if they did not participate. All patients were informed and voluntary consents were obtained in writing.

### Consent for publication

All data has been anonymised and all links between the identity of patients and the datasets shared in this publication were removed.

## Competing interests

MC is associated with Cambridge Precision Medicine Ltd. The other authors declare that they have no competing interests.

## Authors' contributions

RS and MC conceived the experiment, developed and performed the data processing workflow, performed quality control, and drafted the initial manuscript. VMT, IP, CdM, OC, VS developed the consent and ethics protocol, recruited the patients, and performed sampling and extraction of purified DNA from samples. All authors have read and approved of the final manuscript.

## Authors' information

N/A.

## Funding

Limited funding was provided by BGI Genomics UK Co. Ltd.

## Acknowledgements

We would like to thank the patients and their families for their cooperation and contribution to this study.

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
