## [Editor Report]

Editor’s AssessmentMany studies have explored the genetic determinants of COVID-19 severity, these GWAS studies using microarrays or expensive whole-genome sequencing (WGS). Low-coverage WGS data can be imputed using reference panels to enhance resolution and statistical power while maintaining much lower costs, but imputation accuracy is difficult to balance. This work demonstrates how to address these challenges utilising the GLIMPSE1 algorithm, a less resource-intensive tool that produces more accurate imputed data than its predecessors. Generating a dataset containing 79 imputed low-coverage WGS samples from patients with severe COVID-19 symptoms during the initial wave of the SARS-CoV-2 pandemic in Spain. The validation of this imputation and filtering process shows that GLIMPSE1 can be confidently used to impute variants with minor allele frequency up to approximately 2%. After peer review the authors clarified and provided more validation and statistics and figures to help convince this approach was valid. This work showcasing the viability of using low-coverage WGS imputation to generate data for the study of disease-related genetic markers, alongside a validation methodology to ensure the accuracy of the data produced. Helping inspire confidence and encouraging others to deploy similar approaches to other infectious diseases, genetic disorders, or population-based genetic studies. Particularly in large-scale genomic projects and resource-limited settings where sequencing at higher coverage could prove to be prohibitively expensive.Editor’s AssessmentMany studies have explored the genetic determinants of COVID-19 severity, these GWAS studies using microarrays or expensive whole-genome sequencing (WGS). Low-coverage WGS data can be imputed using reference panels to enhance resolution and statistical power while maintaining much lower costs, but imputation accuracy is difficult to balance. This work demonstrates how to address these challenges utilising the GLIMPSE1 algorithm, a less resource-intensive tool that produces more accurate imputed data than its predecessors. Generating a dataset containing 79 imputed low-coverage WGS samples from patients with severe COVID-19 symptoms during the initial wave of the SARS-CoV-2 pandemic in Spain. The validation of this imputation and filtering process shows that GLIMPSE1 can be confidently used to impute variants with minor allele frequency up to approximately 2%. After peer review the authors clarified and provided more validation and statistics and figures to help convince this approach was valid. This work showcasing the viability of using low-coverage WGS imputation to generate data for the study of disease-related genetic markers, alongside a validation methodology to ensure the accuracy of the data produced. Helping inspire confidence and encouraging others to deploy similar approaches to other infectious diseases, genetic disorders, or population-based genetic studies. Particularly in large-scale genomic projects and resource-limited settings where sequencing at higher coverage could prove to be prohibitively expensive.

---

## [Reviewer Report]

Reviewer name and names of any other individual's who aided in reviewer Jong BhakDo you understand and agree to our policy of having open and named reviews, and having your review included with the published papers. (If no, please inform the editor that you cannot review this manuscript.)YesIs the language of sufficient quality?YesPlease add additional comments on language quality to clarify if needed
Are all data available and do they match the descriptions in the paper? YesAdditional CommentsAre the data and metadata consistent with relevant minimum information or reporting standards? See GigaDB checklists for examples <a href="http://gigadb.org/site/guide" target="_blank">http://gigadb.org/site/guide</a>YesAdditional CommentsIs the data acquisition clear, complete and methodologically sound?YesAdditional CommentsIs there sufficient detail in the methods and data-processing steps to allow reproduction?YesAdditional CommentsIs there sufficient data validation and statistical analyses of data quality? YesAdditional CommentsIs the validation suitable for this type of data?YesAdditional CommentsIs there sufficient information for others to reuse this dataset or integrate it with other data?YesAdditional CommentsAny Additional Overall Comments to the AuthorSevere cases of covid-19 patients are critical data. This manuscript deals with detailed clinical information genome set as a subset of exome sequences and provide invaluable data for on-going global covid-19 omics studies. 
RecommendationAccept

---

## [Reviewer Report]

Reviewer name and names of any other individual's who aided in reviewer alfredo iacoangeliDo you understand and agree to our policy of having open and named reviews, and having your review included with the published papers. (If no, please inform the editor that you cannot review this manuscript.)YesIs the language of sufficient quality?YesPlease add additional comments on language quality to clarify if needed
Are all data available and do they match the descriptions in the paper? YesAdditional CommentsAre the data and metadata consistent with relevant minimum information or reporting standards? See GigaDB checklists for examples <a href="http://gigadb.org/site/guide" target="_blank">http://gigadb.org/site/guide</a>YesAdditional CommentsIs the data acquisition clear, complete and methodologically sound?YesAdditional CommentsIs there sufficient detail in the methods and data-processing steps to allow reproduction?YesAdditional CommentsIs there sufficient data validation and statistical analyses of data quality? YesAdditional CommentsIs the validation suitable for this type of data?YesAdditional CommentsIs there sufficient information for others to reuse this dataset or integrate it with other data?YesAdditional CommentsAny Additional Overall Comments to the AuthorThe authors present the release of a new dataset that include low coverage WGS data of 79 individuals who experienced severe covid-19 in Madrid (Spain). The authors processed the data and imputed common variants and they are making this dataset available to the scientific community. They also present the clinical data of these patients in a descriptive and informative fashion. Finally, the authors also validated the quantify of their imputation, showcasing the potential of low coverage WGS as an alternative to microarrays. Overall the manuscript is written very well, clear, and exhaustive. The data is certainly valuable. Its generation and processing and analysis appears robust. Overall I support the publication of this article and dataset. I only have a small number of minor suggestions for the authors:  The sentence "Traditionally, the genotyping process has relied on array technologies as the standard, both at the broader GWAS level and the more specific genetic scoring and genetic diagnostics levels" sounds a little off. I totally understand where the authors come from but given the central role of NGS and Sanger for genetic diagnostics I would suggest the authors to modify accordingly or to keep the GWAS focus.  Please double-check the use a statistical terms in the description of the imputed data. For example: "On average, each VCF file in this rich dataset contains 9.49 million high-confidence single nucleotide variants [95%CI: 9.37 million - 9.61 million] (Figure 1)." The use of CI in this context is a little miss-leading as it is not strictly referring to a distribution of probability but to a finite collection. A range would be more appropriate.  The authors say that they examined the ethnicity of the 79 individuals, however I do not think the ancestry is actually reported anywhere while a few figures show ancestral population data. The authors might clarify or correct the terminology.   Looking at figure 2 the sentence " although the male age distribution exhibits a broader range and higher variability, suggestive of a greater" does not appear justified. The authors might want to clarify or correct accordingly.  The sentence "This exploratory analysis highlights the diverse ways in which severe COVID-19 can present, and the importance of comprehensive and nuanced clinical phenotyping in improving our understanding and management of the disease." suggests some basic clustering might be useful.  The readers might benefit from a couple of graphs or figures quantifying the overlap of the SNPs across samples and maybe one that shows the density of SNPs across the genome. 
RecommendationMinor Revision